# Diversity in HIV epidemic transitions in India: An application of HIV epidemiological metrices and benchmarks

Pradeep Kumar[1]*, Chinmoyee Das[1], Arvind Kumar[1], Damodar Sahu[2], Sanjay K. Rai[3], Sheela Godbole[4], Elangovan Arumugam[5], Lakshmi P. V. M.[6], Shanta Dutta[7], H. Sanayaima Devi[8], Vishnu Vardhana Rao Mendu[2], Shashi Kant[3], Arvind Pandey[2,9], Dandu Chandra Sekhar Reddy[10], Sanjay Mehendale[9,11], Shobini Rajan[1]

1 National AIDS Control Organization, Ministry of Health and Family Welfare, New Delhi, India, 2 Indian Council of Medical Research-National Institute of Medical Statistics, New Delhi, India, 3 All India Institute of Medical Sciences, New Delhi, India, 4 Indian Council of Medical Research-National AIDS Research Institute, Pune, India, 5 Indian Council of Medical Research-National Institute of Epidemiology, Chennai, India, 6 Postgraduate Institute of Medical Education and Research, Chandigarh, India, 7 Indian Council of Medical Research-National Institute of Cholera and Enteric Diseases, Kolkata, India, 8 Regional Institute of Medical Sciences, Imphal, India, 9 Indian Council of Medical Research, New Delhi, India, 10 Institute of Medical Sciences, Banaras Hindu University, Varanasi, India, 11 PD Hinduja Hospital and Medical Research Center, Mumbai, India

* drpradeep2014@gmail.com

## Abstract

### Background

The Joint United Nations Programme on AIDS (UNAIDS) has emphasized on the incidence-prevalence ratio (IPR) and incidence-mortality ratio (IMR) to measure the progress in HIV epidemic control. In this paper, we describe the status of epidemic control in India and in various states in terms of UNAIDS's recommended metrices.

### Method

The National AIDS Control Programme (NACP) of India spearheads work on mathematical modelling to estimate HIV burden based on periodically conducted sentinel surveillance for providing guidance to program implementation and policymaking. Using the results of the latest round of HIV Estimations in 2019, IPR and IMR were calculated.

### Results

National level IPR was 0.029 [0.022–0.037] in 2019 and ranged from 0.01 to 0.15 in various States and Union Territories (UTs). Corresponding Incidence-Mortality Ratio was at 0.881 [0.754–1.014] nationally and ranged between 0.20 and 12.90 across the States/UTs.

### Conclusions

Based on UNAIDS recommended indicators for HIV epidemic control, namely IPR and IMR; national AIDS response in India appears on track. However, the program success is not uniform and significant heterogeneity as well as expanding epidemic was observed at the level

**Data Availability Statement:** The minimal data set underlying the results described in the manuscript can be found in the supplementary tables. Additional de-identified data can be made available

upon request to the researchers who meet the criteria for NACO's data-sharing guidelines. Interested researchers should write to Dr Chinmoyee Das, [c.das@gov.in], Head of Division, Strategic Information (NACO, MoHFW, Govt of India). The guidelines for request are available at http://naco.gov.in/documents/policy-guidelines.

**Funding:** The authors received no specific funding for this work.

**Competing interests:** The authors have declared that no competing interests exist.

of States or UTs. Reinforcing States/UTs specific and focused HIV prevention, testing and treatment initiatives may help in the attainment of 2030 Sustainable Development Goals of ending AIDS as a public health threat by 2030.

## Introduction

The goal of ending the AIDS epidemic by 2030 is integral to attaining the third Sustainable Development Goal (SDG) of ensuring healthy lives and promoting well-being for all at all ages [1]. HIV incidence rate per 1000 uninfected population is the main indicator to measure progress on the HIV/AIDS response in the era of the SDG [2]. While no benchmark for incidence rate has been recommended to be achieved by 2030 for ending the AIDS epidemic, Joint United Nations Programme on HIV/AIDS (UNAIDS) has called to achieve a 90% decline in annual new HIV infections and AIDS-related deaths by 2030 from the baseline value of 2010 [3–5]. Characteristically, between 2010 and 2019, the number of new HIV infections and AIDS-related deaths have declined by 23% and 39% respectively globally [6].

The target of 90% percentage reductions in new HIV infections and AIDS-related deaths as an indicator of progress towards 'ending AIDS as a public health threat' by 2030 has limitations [7, 8]. This indicator does not factor in the epidemic heterogeneity across the globe and ignores the fact that attaining such reductions in low-level epidemic settings would be relatively difficult. Moreover, using the 2010 baseline for these indicators pose a relative disadvantage for the countries having mature interventions who already have achieved strong gains before the baseline year of 2010. In contrast, countries that have scaled up their HIV responses after 2010 have a relative advantage. Further, presenting the progress on new infections and AIDS-related deaths does not adequately depict the association between mortality among people living with HIV (PLHIV), new HIV infections, and the prevalence of HIV making the overall progress on the epidemic.

Recognising these limitations, there have been efforts to search for indicators to measure the progress on the 2030 SDG of 'ending the AIDS epidemic as a public health threat'. Two epidemiological indicators of the sustainability of transmission have been identified as more refined [4, 7–9]. These are: (i) incidence-prevalence ratio (IPR), and (ii) incidence-mortality ratio (IMR). Assuming an average life expectancy of 30 years after a person acquires HIV infection, attainment of a benchmark value of <0.03 on IPR indicates that the PLHIV number will gradually decrease as there are fewer than three new HIV infections per 100 people living with HIV per year and the ending of AIDS epidemic will be achieved. Attainment of benchmark value of <1 for IMR will indicate that the PLHIV number will gradually decrease as there are fewer new infections than deaths. In 2019, UNAIDS reported that 25 countries had achieved the IPR of <0.03 while fewer countries reported achieving desired threshold of IMR [6].

India, with an estimated PLHIV size of 2.35 million and adult prevalence of 0.22% in 2019, is the second-largest HIV/AIDS epidemic in the world [6, 10]. Initiatives of the National AIDS Control Programme (NACP) in the country resulted in a 37% decline in new HIV infections and a 66% decline in AIDS-related deaths between 2010 and 2019. However, no data are available on IPR and IMR from India. In this article, we report the present status of HIV epidemic control in India using the UNAIDS recommended indicators of IPR and IMR at the national and state levels.

## Methods

HIV Surveillance and Estimation is an integral part of the spectrum of activities of the National AIDS Control Organization (NACO) of the Government of India under NACP. This activity is periodically conducted using a specially created programmatic framework built on the foundation of government institutes and partners. As part of the surveillance activity, when primary data and sample collection is done from the survey participants, informed consent is taken from them in alignment with national guidelines. The institutions involved in primary data collection routinely submit their proposals for the surveillance program to their respective ethics committees to seek approval and the survey at each site is initiated only after the local ethics committee approves the proposal. Utilizing this program generated data for policy making and programmatic improvement, including the HIV burden estimation, is a mandate of the NACP in India.

HIV burden estimation is carried out after each round of Surveillance by NACO by making use of aggregated de-identified data. This work has been published periodically in the past. It may be noted that primary data collection from human subjects is not part of HIV burden estimation exercise. The current manuscript is based on the analysis of aggregated de-identified outputs generated through the HIV Estimations 2019 model for each of the Indian State or Union Territory.

We used the modelled estimates for the year 1990 to 2019 from HIV Estimations 2019, the latest round under the NACP, to construct national and state measures of progress on IPR and IMR. The periodic HIV burden estimation exercise under NACP was undertaken employing the UNAIDS supported Spectrum Software (Avenir Health, Glastonbury, Connecticut, USA). The details of the process and method for the same had been described elsewhere [10–17].

In brief, the Spectrum mathematically model demographics, treatment coverage and HIV prevalence data to estimate incidence trend which is first distributed by age, sex and CD4+ counts and then the newly infected population are transitioned over time through age, CD4+ count and treatment (or lack of treatment) categories with death as the final outcome. The 'Uncertainty Analysis' tool in Spectrum generates plausible range of key HIV indicators by running 1000 Monte Carlo iterations combining uncertainty in adult incidence produced by EPP with uncertainty around other key assumptions such as fertilty, incidence, mortality etc based on global or regional values. The tool exports estimates for each of 1000 iteration for key indicators.

In India, the estimation process has been designed to develop sub-national level (State/ Union Territory) models where State/ Union Territory-specific demographic, treatment coverage, prevalence and surveillance data are inputted. Epidemiological parameters such as patterns of incidence, progression, mortality, and fertility derived from scientific studies have been in-built in the Spectrum Software while computing the desired outputs.

The Spectrum software is updated periodically under the guidance of the UNAIDS Reference Group on Estimates, Modeling and Projections. The details of the updates are available on the website of reference group (www.epidem.org) and the software developer website (https://avenirhealth.org/software-spectrum.php). HIV Estimations 2019 under NACP in India was implemented using the 5.80 version of Spectrum Software.

For the current study, we used aggregated deidentified outputs for all States/ Union Territories of India generated through HIV Estimations 2019. This included year-wise data from 1990–2019 on annual new HIV infections and annual total deaths among people living with HIV (PLHIV). Using this data we calculated IPR as the ratio of new HIV infections over PLHIV in a given year for a given geography and IMR as the ratio of new HIV infections over all-cause mortality among PLHIV in a given year for given geography as per standard

definitions [4].

$$IPR = \frac{Number\ of\ new\ infections\ in\ a\ given\ reference\ year\ and\ geography}{Number\ of\ PLHIV\ in\ the\ given\ reference\ year\ and\ geography}$$

$$IMR = \frac{Number\ of\ new\ infections\ in\ a\ given\ reference\ year\ and\ geography}{Number\ of\ deaths\ among\ PLHIV\ in\ the\ given\ reference\ year\ and\ geography}$$

The uncertainty bound for a State/UT for IPR/IMR was estimated using each of the 1000 iteration values of new HIV infections, PLHIV size and total deaths among PLHIV generated through the 'Uncertainty Analysis' tool. After calculating the IPR/IMR for each of the iteration for a State/UT, we obtained the 2.5% and 97.5% percentiles of the 1000 ratios to inform the 95% uncertainty bound estimation for the given State/UT.

To inform the uncertainty bound around national IPR/IMR, we summed the given indicator from all State/UTs by each of 1000 iteration to produce 1000 estimates of incidence, prevalence and mortality for the country. Then we calculated IPR/IMR for each of the 1000 aggregated iterations which were used to generate the uncertainty bound for national estimates.

The current study relied on the analysis of aggregated de-identified outputs generated through the HIV Estimations 2019 model of each State/UTs. As there was no primary data collection for the current analysis, the ethics review was not sought.

## Results

Overall, 2.38 million people were estimated to be living with HIV (PLHIV) in 2019 in India. States of Andhra Pradesh, Karnataka, Tamil Nadu, Telangana (in southern India), Maharashtra, Gujarat (in western India), Punjab (in northern India), Uttar Pradesh (in central India), Bihar, West Bengal (eastern India) are the 10 top-ranking States in India in terms of PLHIV size (Fig 1, S1 Table).

Nationally, the incidence: prevalence ratio was at 0.029 [0.022–0.037] in 2019 while the incidence: mortality ratio was at 0.881 [0.754–1.014] (Table 1, Figs 2 and 3). The IPR had an overall declining trend nationally with estimates of 0.098 [0.076–0.120] in 2000, 0.041 [0.034–0.049] in 2010, and 0.029 [0.022–0.037] in 2019. Nationally, the IMR declined to 0.481 [0.437–0.530] in 2007 and thereafter had a gradual upward trend to 0.569 [0.501–0.668] in 2010 and to 0.881 [0.754–1.014] in 2019. State/ UT-wise IPR and IMR for the period 1990–2019 may be seen at supporting information at S2 Table.

Overall, 21 States/ UTs had IPR of more than 0.03 in 2019. This inlcuded seven States in the north-eastern region of India including Arunachal Pradesh [0.100, 0.061–0.122], Assam [0.059, 0.049–0.073], Meghalaya [0.057, 0.049–0.067], Mizoram [0.062, 0.047–0.075], Nagaland [0.053, 0.044–0.060], Sikkim [0.055, 0.030–0.100] and Tripura [0.153, 0.136–0.169]. IPR of less than 0.03 was noted in five States including Andhra Pradesh [0.009, 0.004–0.017], Karnataka [0.006, 0.003–0.011] and Tamil Nadu [0.016, 0.008–0.024] in southern parts of India. Rest of the States/UTs had IPR with uncertainty overlapping 0.03.

In Andhra Pradesh, IPR declined from 0.125 [0.069–0.186] in 2000 to 0.016 [0.010–0.023] in 2010 to 0.009 [0.004–0.017] in 2019. In Karnataka, IPR declined from 0.125 [0.063–0.183] in 2000 to 0.015 [0.009–0.022] in 2010 to 0.006 [0.003–0.011] in 2019. In Tamil Nadu, IPR declined from 0.054 [0.042–0.118] in 2000 through 0.024 [0.015–0.033] in 2010 to 0.016 [0.008–0.024] in 2019.

IMR was less than 1 in 2019 in seven States including that of Andhra Pradesh [0.199, 0.099–0.338], Karnataka [0.173, 0.088–0.322], Maharasthra [0.664, 0.423–0.894], Tamil Nadu [0.595,

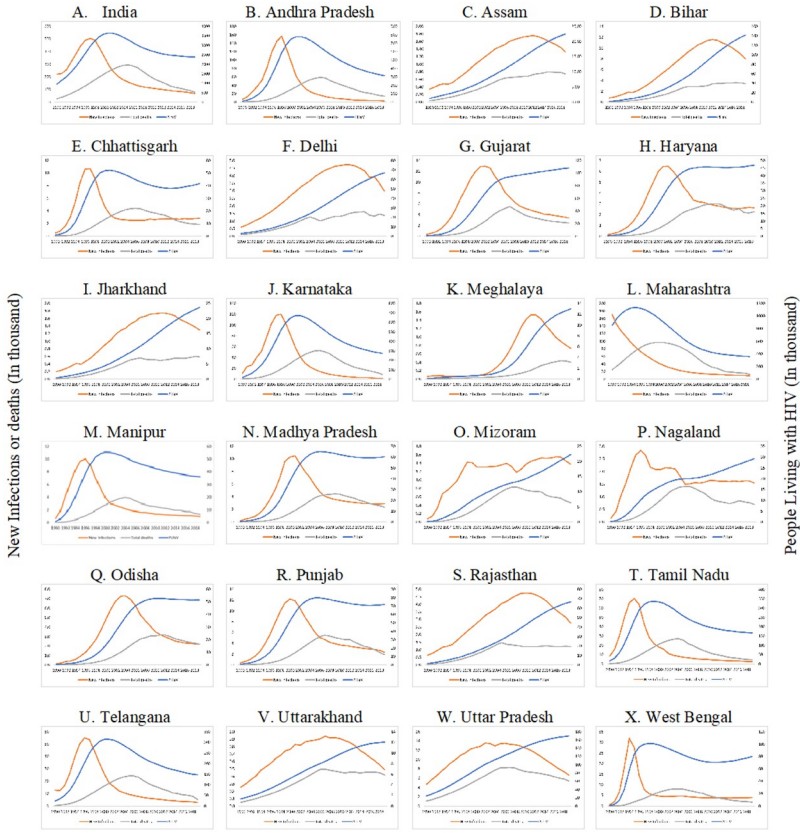

**Fig 1. Annual new HIV infections, annual all-cause mortality among PLHIV and total number of PLHIV by select States in India, 1990–2019.** The years (1990–2019) are reflected on X-axis; number of new infections (in 1000s) and total deaths and PLHIV (in 1000s) are on primary Y-axis while total number of PLHIV (in 1000s) is on secondary Y-axis. Orange line depicts new infections, grey line depicts total deaths among PLHIV, and blue line represents total PLHIV size.

0.315–0.866] and Telangana [0.525, 0.254–0.768]. In Andhra Pradesh, IMR was at 3.104 [1.267–6.115] in 2000, 0.192 [0.119–0.281] in 2010 and 0.199 [0.099–0.338] in 2019. In Karnataka, IMR was at 2.727 [1.043–4.675] in 2000, 0.176 [0.100–0.264] in 2010 and 0.173 [0.088–0.322] in 2019. In Maharasthra, IMR declined to 0.253 [0.165–0.390] in 2006 and then increased to 0.664 [0.423–0.894] in 2019. In Tamil Nadu, IMR was at 0.865 [0.477–3.508] in 2000, 0.281 [0.211–0.365] in 2005, 0.374 [0.228–0.636] in 2010 and 0.595 [0.315–0.866] in 2019. In Telangana, IMR declined to 0.325 [0.223–0.413] in 2007 and was at 0.525 [0.254–0.768] in 2019.

Eighteen States/UTs (Arunachal Pradesh, Assam, Bihar, Chhattisgarh, Delhi, Gujarat, Jharkhand, Jammu & Kashmir, Meghalaya, Mizoram, Nagaland, Rajasthan, Sikkim, Tripura, Uttar Pradesh, West Bengal, Dadra & Nagar Haveli and Daman & Diu) had estimated IMR of more than 1 in 2019. Rest of the States/UTs had IMR with uncertainty overlapping 1 in 2019.

The initial declining trend followed by stabilization or upward trend in IMR, as observed at the national level, was also noted in some of the States/ UTs including Chhattisgarh, Gujarat, Haryana, Manipur, Madhya Pradesh, Mizoram, Nagaland, Odisha, Punjab and West Bengal.

**Table 1. Incidence-prevalence ratio and incidence-mortality ratio (with uncertainty bounds) by States/UTs in India, 2019.**

| State/UT | IPR | | | IMR | | |
|---|---|---|---|---|---|---|
| | Point Estimate | Lower Bound | Upper Bound | Point Estimate | Lower Bound | Upper Bound |
| Andaman & Nicobar Islands | 0.041 | 0.024 | 0.093 | 0.952 | 0.478 | 1.972 |
| Andhra Pradesh | 0.009 | 0.004 | 0.017 | 0.199 | 0.099 | 0.338 |
| Arunachal Pradesh | 0.100 | 0.061 | 0.122 | 3.476 | 1.862 | 4.742 |
| Assam | 0.059 | 0.049 | 0.073 | 1.790 | 1.512 | 2.313 |
| Bihar | 0.057 | 0.037 | 0.070 | 2.358 | 2.003 | 2.930 |
| Chandigarh | 0.056 | 0.029 | 0.074 | 1.180 | 0.681 | 1.546 |
| Chhattisgarh | 0.068 | 0.052 | 0.079 | 1.545 | 1.316 | 1.712 |
| Dadra & Nagar Haveli | 0.108 | 0.059 | 0.127 | 6.727 | 3.376 | 9.474 |
| Daman & Diu | 0.063 | 0.041 | 0.079 | 2.875 | 1.816 | 3.707 |
| Delhi | 0.045 | 0.035 | 0.057 | 2.174 | 1.798 | 2.741 |
| Goa | 0.010 | 0.003 | 0.028 | 0.281 | 0.089 | 0.534 |
| Gujarat | 0.031 | 0.025 | 0.038 | 1.378 | 1.161 | 1.663 |
| Haryana | 0.056 | 0.043 | 0.067 | 1.160 | 0.921 | 1.376 |
| Himachal Pradesh | 0.022 | 0.018 | 0.028 | 1.235 | 0.929 | 1.673 |
| Jammu & Kashmir | 0.048 | 0.028 | 0.091 | 1.831 | 1.304 | 3.672 |
| Jharkhand | 0.055 | 0.040 | 0.071 | 2.192 | 1.783 | 3.027 |
| Karnataka | 0.006 | 0.003 | 0.011 | 0.173 | 0.088 | 0.322 |
| Kerala | 0.029 | 0.020 | 0.043 | 1.166 | 0.862 | 1.485 |
| Madhya Pradesh | 0.048 | 0.033 | 0.061 | 1.225 | 0.920 | 1.420 |
| Maharashtra | 0.024 | 0.014 | 0.040 | 0.664 | 0.423 | 0.894 |
| Manipur | 0.027 | 0.017 | 0.040 | 0.733 | 0.472 | 0.979 |
| Meghalaya | 0.057 | 0.049 | 0.067 | 1.821 | 1.521 | 2.450 |
| Mizoram | 0.062 | 0.047 | 0.075 | 2.972 | 2.245 | 3.841 |
| Nagaland | 0.053 | 0.044 | 0.060 | 2.216 | 1.942 | 2.579 |
| Odisha | 0.042 | 0.034 | 0.051 | 0.997 | 0.851 | 1.187 |
| Puducherry | 0.070 | 0.043 | 0.103 | 0.881 | 0.589 | 1.095 |
| Punjab | 0.033 | 0.024 | 0.045 | 1.234 | 0.934 | 1.563 |
| Rajasthan | 0.042 | 0.031 | 0.050 | 2.255 | 1.882 | 2.746 |
| Sikkim | 0.055 | 0.030 | 0.100 | 3.500 | 1.566 | 5.144 |
| Tamil Nadu | 0.016 | 0.008 | 0.024 | 0.595 | 0.315 | 0.866 |
| Telangana | 0.019 | 0.009 | 0.031 | 0.525 | 0.254 | 0.768 |
| Tripura | 0.153 | 0.136 | 0.169 | 12.914 | 9.612 | 16.666 |
| Uttar Pradesh | 0.040 | 0.030 | 0.049 | 1.238 | 1.042 | 1.503 |
| Uttarakhand | 0.041 | 0.032 | 0.052 | 1.167 | 0.952 | 1.554 |
| West Bengal | 0.050 | 0.034 | 0.064 | 2.110 | 1.738 | 2.545 |
| India | 0.029 | 0.022 | 0.037 | 0.881 | 0.754 | 1.014 |

## Discussions

The use of epidemiological metrices to assess trajectories of the HIV/AIDS epidemic and high-light areas for intervention is not new. The incidence:prevalence ratio has been used in the United Kingdom, United States of America, Denmark, Norway and Sweden in the past to estimate the HIV transmission dynamics [18–20]. These dynamic metrics are increasingly used since 2015 to assess the status of HIV epidemic control because they are rooted in sound epidemiological principles and analysis [4, 7–9, 21, 22]. These are very useful metrics to assess whether the direction of response to the epidemic is 'on-track' in the context of reducing the

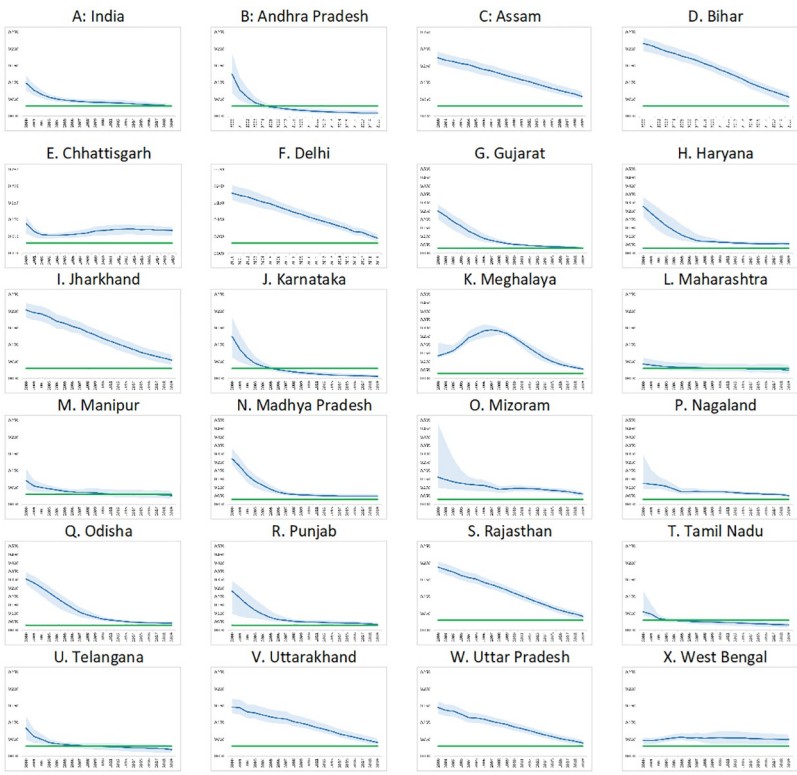

**Fig 2. Incidence-prevalence ratio by States in India, 2000–2019.** The years (2000–2019) are reflected on X-axis while IPR is on Y-axis. The highest bound on Y-axis is 0.250 for all except for Gujarat, Haryana, Madhya Pradesh, Meghalaya, Mizoram, Nagaland, Odisha and Punjab (0.500). Green line depicts target values of IPR for epidemic control, blue line represents point estimate and light blue shaded areas represents the uncertainty bounds for IPR of India/State for the period 2000–2019.

size of the epidemic and whether the epidemic is downsizing. However, the application of IPR and IMR to measure the Indian HIV epidemic transition has not been demonstrated earlier. This paper analyses the data from HIV estimations 2019 in India by its State/ UTs and presents the level and trends of IPR and IMR at the national and sub-national levels.

The significant decline in annual new HIV infections and AIDS-related deaths with continued low adult HIV prevalence has been documented in India [11, 13]. With a declining trend in IPR nationallly, the current analysis corroborates that HIV/AIDS epidemic control in India is on track. Yet, with an IPR of 0.029 [0.022–0.037] in 2019, programme intensity need to be maintained and augmented to attain and sustain the target IPR of <0.03 conclusively to achieve the epidemic control.

Nationally, the incidencce: mortality ratio in India is increasing since 2007. Given the trend, the IMR may surpass the threshold value of 1 in near future and the overall size of the people living with HIV/AIDS in India may also increase. This is consistent with a strong scale-up in the uptake of antiretroviral therapy (ART) leading to a very rapid decline in annual AIDS-related mortality [23].

In three southern states (Andhra Pradesh, karnataka and Tamil Nadu) both IPR and IMR is conclusively less than the target threshold in 2019. These three are among the states where historically the HIV epidemic was much higher than in the rest of India and hence the national AIDS response has focussed on these states since the early days of initiation of NACP with 70% or more of the estimated PLHIV are already on ART [23–26]. The response to the HIV

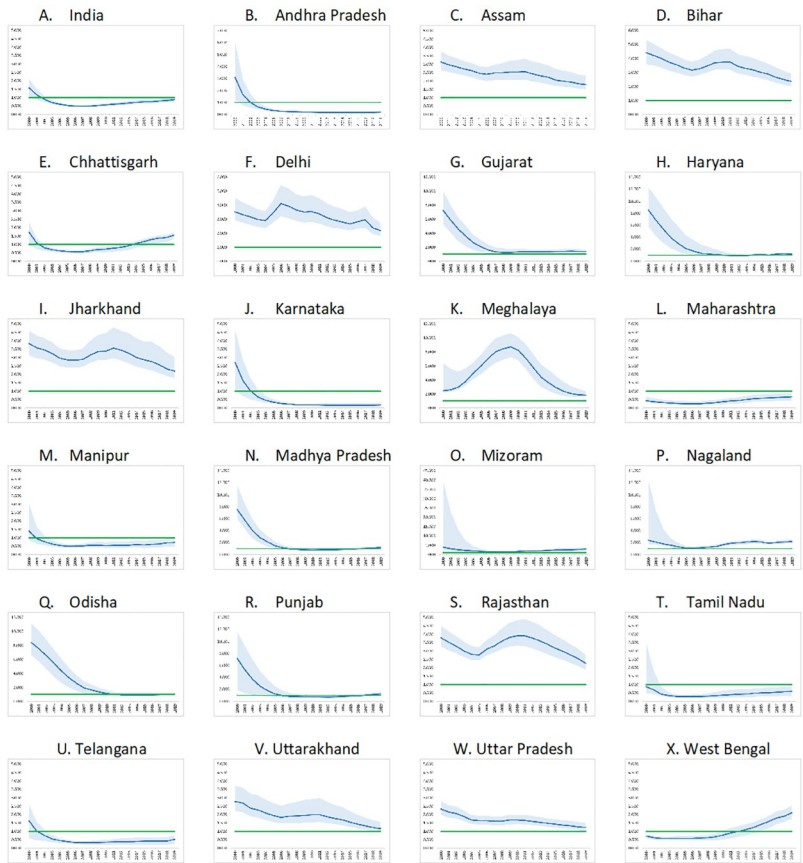

**Fig 3. Incidence-mortality ratio by States in India, 2000–2019.** The years (2000–2019) are reflected on X-axis while IMR is on Y-axis. The highest bound on Y-axis is 5.000 for all except for Andhra Pradesh (7.000), Bihar (6.000), Delhi (6.000), Gujarat (12.000), Haryana (14.000), Meghalaya (12.000), Madhya Pradesh (14.000), Mizoram (45.000), Nagaland (14.000), Odisha (12.000) and Punjab (14.000). Green line depicts target value of IMR for epidemic control, blue line represents point estimate and light blue shaded areas represents the uncertainty bounds for IMR of India/ State for the period 2000–2019.

epidemic in these states is on track and the total size of the PLHIV is expected to continue to decline in near future.

Twenty-one States/UTs across the country had IPR of more than 0.03 in 2019. Among these, sixteen (including seven States of Arunachal Pradesh, Assam, Meghalaya, Mizoram, Nagaland, Sikkim and Tripura in north-eastern India) also had IMR of more than 1 in 2019. These States need to be focussed under the programme as the PLHIV size in these States/UTs will grow over time with epidemic not in a state of control.

The current progress on the HIV epidemic transition threshold in India is consistent with the progress seen globally. The global incidence: prevalence ratio was 0.04 in 2019 with 25 countries having achieved the milestone of 0.03. Similarly, the threshold of <1 for IMR is expected to be achieved by a few countries with large HIV epidemics [27].

Both IPR and IMR are dynamic measures with roots in epidemiological theory about the sustainability of the transmission, but still have certain limitations [4–8, 23, 28, 29]. IPR threshold value of 0.03 for epidemic control assumes average survival of 33 years after HIV infection. The average survival may differ by regions and country based on the ART uptake, adherence etc which will have an impact on the IPR threshold value. The IPR threshold value to be

asssumed for epidemic control in the context of ART uptake may be one of the factors while deliberating the threshold and behaviour of IPR [4, 7].

IPR and IMR work well when applied to the national or state population as a whole but are not suitable for the population subgroups where HIV acquisition and transmission is not be limited within the index subgroup. In India, where the HIV epidemic is concentrated with HIV prevalence among female sex workers, men who have sex with men, transgender people, and people who inject drugs are 7–28 times that of overall adult prevalence, this may lead to a false sense of complacency. Also, IMR may go down below the threshold level of 1 in settings with high AIDS-related mortality, as indicated in the case of states like Manipur, and thus be fallacious if seen in isolation in locations with low ART coverage.

Further, the scale-up of ART therapy reduces mortality, which shrinks the denominator and may create an upward trend in IMR even if incidence is decreasing and epidemic response is on track. The upward trend in IMR after 2007, seen at the country-level in India, is a resultant of this dynamics. Further, these measures do not reflect the status of legal, policy and social enablers and thus ignore the critical structural issues in AIDS response. Still, these two ratios provide critical insights into the current and future status of the epidemic in geographies as a whole by establishing if the epidemic is expanding or shrinking.

Our study limitations are also from source data. The study has used modelled estimates. There are inherent limitations to estimations based on modelling [4, 30]. The quality of modelled estimates depends on the quality of empirical data used as input data to inform the programme coverage and prevalence. The wider uncertainty bounds in 2019 estimates had limited capacity of this analysis to draw conclusive inference. The uncertainty bounds of estimates are usually influenced by aspects like quantity of surveillance data and use of population-based survey. Overall, the quanity and quality of sentinel surveillance in India has been described as good [31]. Also, prevalence estimates from population-based survey have been used in India for HIV burden estimation. The wider uncertainty bounds may be the outcome of the fact the State/UT-wise model is prepared during the HIV burden estimation process in India. Still, there is a need for examining the wider uncertainty bounds noted in modelled estimates.

The current study is based on analysis of outputs of a mathematical modeeling process recommended by UNAIDS. While modelled estimates on various epidemiological indicatiors like prevention of new HIV infections and AIDS-related mortality using globally used model are accepeted under NACP, there is a need for validation of modelled estimates of incidence and mortality. Investment in the components of second-generation surveillance focussing on incidence, mortality and case-based surveillance will triangulate the modelled estimates vis-à-vis empirical evidence and finally lead to a more reliable assessment on progress on the HIV epidemic transition threshold.

Despite the limitations, our study, the first to quantify the progress on epidemiological metrices by State and UTs in India to the best of our knowledge, highlighting potential challenges for national AIDS response. The overall progress at the national level masks the subnational heterogeneity where the epidemic is expanding. To ensure that epidemic control is truly realised as envisioned in the 2030 SDGs, the interventions need to be tailored, expanded and intensified in the Indian States/ UTs, especially on prevention aspects, to reach the defined metric's benchmark corresponding to epidemic transition.

## Supporting information

**S1 Table. Annual new HIV infections, annual all-cause mortality among PLHIV and total number of PLHIV (in 100,000) by States/UTs in India, 1990–2019.**
(PDF)

**S2 Table. IPR and IMR by States/UT in India, 1990–2019.**
(PDF)

## Acknowledgments

The project was part of the Surveillance and Epidemiological activities of the National AIDS Control Programme of India. The authors thank the Project Directors and Strategic Information Team of all State AIDS Control Societies for their support in undertaking HIV Surveillance and Estimation activities in their states. We thank academic editor and reviewer for insightful comments and suggestions on the manuscript. We thank Keith Sabin (UNAIDS Geneva) for the techhnical support in generating uncertainty bounds.

## Author Contributions

**Conceptualization:** Pradeep Kumar, Shobini Rajan.

**Data curation:** Pradeep Kumar, Arvind Kumar, Damodar Sahu.

**Formal analysis:** Pradeep Kumar.

**Methodology:** Pradeep Kumar.

**Project administration:** Pradeep Kumar, Damodar Sahu, Vishnu Vardhana Rao Mendu, Shobini Rajan.

**Writing – original draft:** Pradeep Kumar.

**Writing – review & editing:** Chinmoyee Das, Sanjay K. Rai, Sheela Godbole, Elangovan Arumugam, Lakshmi P. V. M., Shanta Dutta, H. Sanayaima Devi, Shashi Kant, Arvind Pandey, Dandu Chandra Sekhar Reddy, Sanjay Mehendale, Shobini Rajan.

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
