## [Decision Letter · Decision Letter 0]

9 Feb 2022

PONE-D-21-34403Diversity in HIV Epidemic transitions in India: An application of HIV epidemiological metrices and benchmarksPLOS ONE

Dear Dr. Kumar,

Thank you for submitting your manuscript to PLOS ONE. After careful consideration, we feel that it has merit but does not fully meet PLOS ONE’s publication criteria as it currently stands. Therefore, we invite you to submit a revised version of the manuscript that addresses the points raised during the review process.

We look forward to receiving your revised manuscript.

Kind regards,

Bharat S. Parekh, Ph.D.

Academic Editor

PLOS ONE

Journal Requirements:

2. You indicated that ethical approval was not necessary for your study. We understand that the framework for ethical oversight requirements for studies of this type may differ depending on the setting and we would appreciate some further clarification regarding your research. Could you please provide further details on why your study is exempt from the need for approval and confirmation from your institutional review board or research ethics committee (e.g., in the form of a letter or email correspondence) that ethics review was not necessary for this study? Please include a copy of the correspondence as an ""Other"" file.

3. Please amend your authorship list in your manuscript file to include all author

Reviewers' comments:

Reviewer's Responses to Questions

**Comments to the Author**

1. Is the manuscript technically sound, and do the data support the conclusions?

Reviewer #1: Yes

2. Has the statistical analysis been performed appropriately and rigorously? 

Reviewer #1: Yes

3. Have the authors made all data underlying the findings in their manuscript fully available?

Reviewer #1: Yes

4. Is the manuscript presented in an intelligible fashion and written in standard English?

Reviewer #1: Yes

5. Review Comments to the Author

Reviewer #1: I’d like to thank the authors for the opportunity to review their manuscript. This is an important topic that is well explored by the paper. I recommend it for publication with a few suggested improvements/comments.

When discussing disease control, the ideal metric to be monitored is Rt. IPR and IMR are important proxies that are helpful in measuring the growth or reduction of disease in a population. I think the authors do a good job of explaining why we should care about these metrics and how they relate to epidemic control. However, I think it would be useful to add information about their limitations.

1.a. The relation between IPR, epidemic control and the .03 target assumes that secondary infections are equally likely to occur at any time during the (on average) 30 years of time an individual is infected. While a useful assumption, this is in general not the case. Scale up of test and treat has reduced the infectivity of diagnosed individuals so that long term infections are less likely to transmit than short term ones. The result is that epidemic control may be reached at much higher than .03 IPR.

1.b. For IMR, changing time between infection and death can create trends that have little to do with epidemic control. I believe you see this in the India data. The scale up of ART therapy reduces mortality, which shrinks the denominator. This can create an upward trend in IMR even if incidence is decreasing. I interpret the upward trend after 2005 as an artifact of this.

Other minor points:

If you can, it would be great if the images could be as high of quality as possible. I found myself zooming in on the individual graphs and they were readable, but could get a bit pixelated.

95-103: These equations are unreadable. It took me a while to realize that they were fractions. I thought they were formatting errors.

164: This is too speculative. It may happen and the data may support continued reduction, but we shouldn’t just expect that HIV will go away.

Grammar:

Consider doing a close read through for grammar. Here are a few things I noticed.

7: remove second “on”

43: doesnot —> does not

43: add “in” after “factor”

59: delete “will”

68: delete “.”

69: Method —> Methods

71: states —> state

119: I don’t see supplementals. Did you mean figures?

6. PLOS authors have the option to publish the peer review history of their article (what does this mean?). If published, this will include your full peer review and any attached files.

Reviewer #1: No

---

## [Author Response · Author response to Decision Letter 0]

20 Apr 2022

Response to the observations of reviewers and suggestions of editors in the context of the manuscript ID “[PONE-D-21-34403] - [EMID:49e8df8c235ee894]” titled "Diversity in HIV Epidemic transitions in India: An application of HIV epidemiological metrices and benchmarks"

***

Response to the comments from Reviewer and Editor

We have gone through the observations of the reviewer and editor for our manuscript ID “[PONE-D-21-34403] - [EMID:49e8df8c235ee894]” titled "Diversity in HIV Epidemic transitions in India: An application of HIV epidemiological metrices and benchmarks". 

We thank the reviewers for their constructive comments and we believe that we are now able to provide more clarity to the readers. 

We have provided a detailed point-by-point response to all observations/comments (reviewer’s/editor’s comments in black, our replies in blue). Line numbering refers to the revised manuscript. 

Reviewer’s Observations/Comments and Authors’ Response

1. General Observation/Comment

I’d like to thank the authors for the opportunity to review their manuscript. This is an important topic that is well explored by the paper. I recommend it for publication with a few suggested improvements/comments.

Reply: We sincerely thank the reviewer for appreciation of our work. 

2. Technical Comments

(i) When discussing disease control, the ideal metric to be monitored is Rt. IPR and IMR are important proxies that help measure the growth or reduction of disease in a population. I think the authors do a good job of explaining why we should care about these metrics and how they relate to epidemic control. However, I think it would be useful to add information about their limitations.

Reply: We thank the reviewer for a clear guidance. We have added the limitations in the revised manuscript [line number ‘244’ to ‘266’]. 

(ii) The relation between IPR, epidemic control and the .03 target assumes that secondary infections are equally likely to occur at any time during the (on average) 30 years of time an individual is infected. While a useful assumption, this is in general not the case. Scale up of test and treat has reduced the infectivity of diagnosed individuals so that long term infections are less likely to transmit than short term ones. The result is that epidemic control may be reached at much higher than .03 IPR.

Reply: We agree with the reviewer. We have added this context in the revised manuscript [lines 244-251]. 

(iii) For IMR, changing the time between infection and death can create trends that have little to do with epidemic control. I believe you see this in the India data. The scale-up of ART therapy reduces mortality, which shrinks the denominator. This can create an upward trend in IMR even if incidence is decreasing. I interpret the upward trend after 2005 as an artifact of this.

Reply: We thank the reviewer for this insight. Although we had alluded to this point in paragraph 2 of the discussion section in our original manuscript, we have now included this additional perspective in the limitations of IMR in high ART coverage settings [lines 260-266 of the revised manuscript].

3. Other minor points

(a) If you can, it would be great if the images could be as high of quality as possible. I found myself zooming in on the individual graphs and they were readable, but could get a bit pixelated.

Reply: In pursuance of the suggestion of the reviewer, we have now updated the legends of the graphs clearly stating which colour of the line graph represents which indicator for the ease of understanding of the reader. The legend has been also updated to reflect the items on the X and Y-axis. 

(b) 95-103: These equations are unreadable. It took me a while to realize that they were fractions. I thought they were formatting errors.

Reply: We thank the reviewer for this valuable suggestion. We have now presented the equations in an improved format by using the insert equation function. The revised equations are on lines 138 and 140. 

(c) 164: This is too speculative. It may happen and the data may support continued reduction, but we shouldn’t just expect that HIV will go away.

Reply: We fully agree with the reviewer. We have edited the statement appropriately [line 217 of the revised manuscript]. 

(d) 119: I don’t see supplementals. Did you mean figures?

Reply: We had added the supplement at the time of original submission, but perhaps it was not accessible. We have uploaded the supplement again. 

4. Grammer

Consider doing a close read through for grammar. Here are a few things I noticed.

a. 7: remove second “on”

b. 43: doesnot —> does not

c. 43: add “in” after “factor”

d. 59: delete “will”

e. 68: delete “.”

f. 69: Method —> Methods

g. 71: states —> state

Reply: We thank the reviewer and made changes as suggested. We have carefully reviewed our revised manuscript. 

1. General Observation/Comment

I’d like to thank the authors for the opportunity to review their manuscript. This is an important topic that is well explored by the paper. I recommend it for publication with a few suggested improvements/comments.

Reply: We sincerely thank the reviewer for appreciation of our work. 

2. Technical Comments

(i) When discussing disease control, the ideal metric to be monitored is Rt. IPR and IMR are important proxies that help measure the growth or reduction of disease in a population. I think the authors do a good job of explaining why we should care about these metrics and how they relate to epidemic control. However, I think it would be useful to add information about their limitations.

Reply: We thank the reviewer for a clear guidance. We have added the limitations in the revised manuscript [line number ‘244’ to ‘266’]. 

(ii) The relation between IPR, epidemic control and the .03 target assumes that secondary infections are equally likely to occur at any time during the (on average) 30 years of time an individual is infected. While a useful assumption, this is in general not the case. Scale up of test and treat has reduced the infectivity of diagnosed individuals so that long term infections are less likely to transmit than short term ones. The result is that epidemic control may be reached at much higher than .03 IPR.

Reply: We agree with the reviewer. We have added this context in the revised manuscript [lines 244-251]. 

(iii) For IMR, changing the time between infection and death can create trends that have little to do with epidemic control. I believe you see this in the India data. The scale-up of ART therapy reduces mortality, which shrinks the denominator. This can create an upward trend in IMR even if incidence is decreasing. I interpret the upward trend after 2005 as an artifact of this.

Reply: We thank the reviewer for this insight. Although we had alluded to this point in paragraph 2 of the discussion section in our original manuscript, we have now included this additional perspective in the limitations of IMR in high ART coverage settings [lines 260-266 of the revised manuscript].

3. Other minor points

(a) If you can, it would be great if the images could be as high of quality as possible. I found myself zooming in on the individual graphs and they were readable, but could get a bit pixelated.

Reply: In pursuance of the suggestion of the reviewer, we have now updated the legends of the graphs clearly stating which colour of the line graph represents which indicator for the ease of understanding of the reader. The legend has been also updated to reflect the items on the X and Y-axis. 

(b) 95-103: These equations are unreadable. It took me a while to realize that they were fractions. I thought they were formatting errors.

Reply: We thank the reviewer for this valuable suggestion. We have now presented the equations in an improved format by using the insert equation function. The revised equations are on lines 138 and 140. 

(c) 164: This is too speculative. It may happen and the data may support continued reduction, but we shouldn’t just expect that HIV will go away.

Reply: We fully agree with the reviewer. We have edited the statement appropriately [line 217 of the revised manuscript]. 

(d) 119: I don’t see supplementals. Did you mean figures?

Reply: We had added the supplement at the time of original submission, but perhaps it was not accessible. We have uploaded the supplement again. 

4. Grammer

Consider doing a close read through for grammar. Here are a few things I noticed.

a. 7: remove second “on”

b. 43: doesnot —> does not

c. 43: add “in” after “factor”

d. 59: delete “will”

e. 68: delete “.”

f. 69: Method —> Methods

g. 71: states —> state

Reply: We thank the reviewer and made changes as suggested. We have carefully reviewed our revised manuscript. 

Editor’s Observations/Comments and Authors’ Response 

Reply: We thank the editor for the suggestion. We have updated the manuscript to meet PLOS ONE's style requirements to the best of our understanding. 

2. You indicated that ethical approval was not necessary for your study. We understand that the framework for ethical oversight requirements for studies of this type may differ depending on the setting and we would appreciate some further clarification regarding your research. Could you please provide further details on why your study is exempt from the need for approval and confirmation from your institutional review board or research ethics committee (e.g., in the form of a letter or email correspondence) that ethics review was not necessary for this study? Please include a copy of the correspondence as an ""Other"" file.

Reply: We thank the editor for the observation. We wish to clarify that HIV Surveillance and Estimation is an integral part of the spectrum of activities of the National AIDS Control Organization (NACO) of the Government of India. This activity is periodically conducted using a specially created programmatic framework built on the foundation of government institutes and partners. As part of the surveillance activity, when primary data and sample collection is done from the survey participants, informed consent is taken from them in alignment with national guidelines. The institutions involved in primary data collection routinely submit their proposals for the surveillance program to their respective ethics committees to seek approval and the survey at each site is initiated only after the local ethics committee approves the proposal. However, utilizing this program generated data for policy making and programmatic improvement is a mandate of the National AIDS Control Program. We have attached the Ethics Committee approval for the activity of integrated bio-behavioural surveillance and HIV sentinel surveillance for reference as “Other” file. 

HIV burden estimation is carried out after each round of Surveillance by NACO by making use of aggregated de-identified data. This work has been published periodically in the past. It may be noted that primary data collection from human subjects is not part of HIV burden estimation exercise. The current manuscript is based on the analysis of aggregated de-identified outputs generated through the HIV Estimations 2019 model for each of the Indian State or Union Territory. 

In summary, ethics committee approvals have been taken by all the agencies that contribute primary data to HIV surveillance program, but state level HIV burden estimations have been done as part of mandate of the National AIDS Control Program.

3. Please amend your authorship list in your manuscript file to include all author

Reply: We thank the editor for the suggestion. We have updated the manuscript file to include all author [line 4-19]. 

4. In your Data Availability statement, you have not specified where the minimal data set underlying the results described in your manuscript can be found. PLOS defines a study's minimal data set as the underlying data used to reach the conclusions drawn in the manuscript and any additional data required to replicate the reported study findings in their entirety. All PLOS journals require that the minimal data set be made fully available.

Reply: The minimal data set underlying the results described in the manuscript can be found in the supplementary tables. Additional de-identified data can be made available upon request to the researchers who meet the criteria for NACO’s data-sharing guidelines. Interested researchers should write to Dr Chinmoyee Das, [c.das@gov.in], Head of Division, Strategic Information (NACO, MoHFW, Govt of India). 

Reply: We have reviewed the references and ensured that they have been correctly cited. To the best of our knowledge, we have not cited any papers that have been retracted.

---

## [Decision Letter · Decision Letter 1]

23 May 2022

PONE-D-21-34403R1Diversity in HIV Epidemic transitions in India: An application of HIV epidemiological metrices and benchmarksPLOS ONE

Dear Dr. Kumar,

Thank you for submitting your manuscript to PLOS ONE. After careful consideration, we feel that it has merit but does not fully meet PLOS ONE’s publication criteria as it currently stands. Therefore, we invite you to submit a revised version of the manuscript that addresses the points raised during the review process. In particular, I would like you to address two key points: 1) The data source used, 'India HIV Estimates 2019' shows very large amounts of uncertainty in the estimates of numbers of PLHIV, HIV incidence, and HIV mortality. This uncertainty needs to be carried through into the estimates presented in the manuscript, as the results cannot be correctly interpreted without confidence intervals. The level of uncertainty in the results should also be reflected in the interpretation of the results and the language used throughout the manuscript. For instance, statements such as "Ten States/ UTs of Andaman ... had estimated IMR of one or less." should not be made if the confidence intervals overlap one. 2) The new paragraph starting from line 244 is not correct. I appreciate that it was added in response to a suggestion made by a reviewer, but it is not necessary to make changes suggested by reviewers if they are not factually correct. The prevalence of HIV will grow in a population if the rate of new infections is higher than the rate at which people are removed from the pool of prevalence infections. The time of transmission relative to the duration of disease does not effect the interpretation of the IPR in such a simplistic way. I think the mistake the reviewer was making was in thinking that if people mostly only transmit in the first 5 years following infection, the the IPR at which control is achieved should be 1/5, not 1/30. What that misses is that in that case only prevalent infections in the first 5 years following infection should be included in the numerator for the IPR, and therefore the two factors will cancel out.

We look forward to receiving your revised manuscript.

Kind regards,

Nicky McCreesh

Academic Editor

PLOS ONE

Reviewers' comments:

Reviewer's Responses to Questions

**Comments to the Author**

1. If the authors have adequately addressed your comments raised in a previous round of review and you feel that this manuscript is now acceptable for publication, you may indicate that here to bypass the “Comments to the Author” section, enter your conflict of interest statement in the “Confidential to Editor” section, and submit your "Accept" recommendation.

Reviewer #2: All comments have been addressed

2. Is the manuscript technically sound, and do the data support the conclusions?

Reviewer #2: Yes

3. Has the statistical analysis been performed appropriately and rigorously? 

Reviewer #2: Yes

4. Have the authors made all data underlying the findings in their manuscript fully available?

Reviewer #2: Yes

5. Is the manuscript presented in an intelligible fashion and written in standard English?

Reviewer #2: Yes

6. Review Comments to the Author

Reviewer #2: (No Response)

7. PLOS authors have the option to publish the peer review history of their article (what does this mean?). If published, this will include your full peer review and any attached files.

Reviewer #2: **Yes: **John Stover

---

## [Author Response · Author response to Decision Letter 1]

5 Jun 2022

Academic Editor Observations/Comments and Authors’ Response

1. Comment 1: 

The data source used, 'India HIV Estimates 2019' shows very large amounts of uncertainty in the estimates of numbers of PLHIV, HIV incidence, and HIV mortality. This uncertainty needs to be carried through into the estimates presented in the manuscript, as the results cannot be correctly interpreted without confidence intervals. The level of uncertainty in the results should also be reflected in the interpretation of the results and the language used throughout the manuscript. For instance, statements such as "Ten States/ UTs of Andaman ... had estimated IMR of one or less." should not be made if the confidence intervals overlap one. 

Reply: 

We agree that we have provided only the point estimates for IPR and IMR. We concur with the Editor observations of providing uncertainty bounds. Adding uncertainty bounds on IPR and IMR will not only increase the scientific rigour of the manuscript but also improve inferences as noted by the Editor. 

The current manuscript is based on the outputs generated using Spectrum Model under 2019 round of HIV burden estimates in India. The Spectrum model calculates uncertainty bounds around each estimate including that on new infections and deaths among PLHIV. However, we were of the opinion that using lower bounds or upper bounds of new infections, deaths and PLHIV to create uncertainty bound around IPR and IMR may not be the right approach as these would be applied for both numerator and denominator. 

We noted that UNAIDS publications have provided IPR with uncertainty bounds though method for the same was not described. And hence we reached to Dr Keith Sabin (Strategic Information Department, UNAIDS, Geneva, Switzerland) to understand the calculation of uncertainty bounds. Dr Sabin informed that proportional uncertainty around the new infections is used to calculate the uncertainty bound around IPR and IMR. To be consistent with globally used methodology, we applied the same for our manuscript. This method has been stated in Line Number 136-138 of the revised manuscript. 

The Results section has been updated to include the uncertainty bounds around tables and figures. The table 1, beginning at Line Number 177, now present State/UT-wise IPR and IMR with uncertainty bounds. Besides all the relevant paragraphs (Line Number 159-165, Line Number 180-200) in the results sections have results with uncertainty bounds. 

We have revised the inferences and language of the discussions in view of the insights provided by uncertainty bounds. The revision is reflected in Line Number 208 to 239 of the revised manuscript. 

We have noted the wider uncertainty bounds as one of the limitations (Line Number 270-277).

2. Comment 2: 

The new paragraph starting from line 244 is not correct. I appreciate that it was added in response to a suggestion made by a reviewer, but it is not necessary to make changes suggested by reviewers if they are not factually correct. The prevalence of HIV will grow in a population if the rate of new infections is higher than the rate at which people are removed from the pool of prevalence infections. The time of transmission relative to the duration of disease does not effect the interpretation of the IPR in such a simplistic way. I think the mistake the reviewer was making was in thinking that if people mostly only transmit in the first 5 years following infection, the the IPR at which control is achieved should be 1/5, not 1/30. What that misses is that in that case only prevalent infections in the first 5 years following infection should be included in the numerator for the IPR, and therefore the two factors will cancel out.

Reply: 

We do see the point mentioned by the academic editor. In fact, we reviewed the UNAIDS report titled ‘MAKING THE END OF AIDS REAL: CONSENSUS BUILDING AROUND WHAT WE MEAN BY EPIDEMIC CONTROL’ and paper titled "Epidemiological metrics and benchmarks for a transition in the HIV epidemic’ by Ghys, Peter D., et al. We noted that both of these refers to threshold of 0.03 as more of a ‘rule of thumb’ and indicated for more discussions around the threshold. As ART uptake has significant impact on transmission of new infections, we believe that this is area that may be included as one of the points in future discussions on threshold value and behaviour of IPR.

In view of above, we have retained the reference to potential impact of ART on IPR threshold. However, we have removed the sentence ‘As a result, the epidemic control may be reached at much higher than 0.03 IPR’ to avoid any comments on the threshold level of epidemic control in our manuscript. We have added a sentence ‘Impact of ART uptake on the IPR may be one of the factors while deliberating the threshold and behaviour of IPR’ (Line Number

---

## [Editor Report · Decision Letter 2]

7 Jun 2022

PONE-D-21-34403R2Diversity in HIV Epidemic transitions in India: An application of HIV epidemiological metrices and benchmarksPLOS ONE

Dear Dr. Kumar,

Thank you for submitting your manuscript to PLOS ONE. After careful consideration, we feel that it has merit but does not fully meet PLOS ONE’s publication criteria as it currently stands. Therefore, we invite you to submit a revised version of the manuscript that addresses the points raised during the review process.

Thanks for adding in uncertainty estimates, they greatly increase the potential usefulness and applicability of the work. Unfortunately, the way they are calculated is not quite correct (based on the way it is described in the methods). It takes into account the uncertainty resulting from the uncertainty in the incidence estimates, but not the uncertainty resulting from the mortality and prevalence estimates. This could mean that the uncertainty is underestimated. Alternatively, if the incidence and mortality/prevalence estimates are correlated between model runs, it may mean that the uncertainty is overestimated.

I have not used the Spectrum models myself. Is it possible to export the estimates from individual model runs, using the uncertainty analysis tool? If it is possible, the simplest way to obtain better estimates of the uncertainty would be to divide the incidence by the mortality/prevalence estimate separately for each individual model run, and then to obtain the 2.5% and 97.5% percentiles for the ratios to give 95% uncertainty intervals. Please make sure that the uncertainty intervals that this approach generates seem reasonable to you however, it is possible that the way that the Spectrum model is parameterised and calibrated would mean that this approach would generate unreasonably narrow uncertainty intervals.

Is this is not possible, then there may not be any better approach than the one used, but please discuss the limitations of the way that the uncertainty intervals are generated in the Discussion section.

I agree entirely that the threshold of 0.03 is a rule of thumb only, and feel that a paragraph discussing this is important. What I do not understand is this sentence: “the 0.03 target assumes that secondary infections are equally likely to occur at any time during the (on average) 30 years of time an individual is infected”. As far as I can see, the references you provided in your response say nothing about the target assuming this, and I do not understand how it does. Please explain this statement and/or provide a reference for it, or remove it. It could be replaced with a brief discussion of the major assumption made in setting the threshold: that people live with HIV for a mean of 30 years.

We look forward to receiving your revised manuscript.

Kind regards,

Nicky McCreesh

Academic Editor

PLOS ONE
---

## [Author Response · Author response to Decision Letter 2]

18 Jun 2022

PONE-D-21-34403R2

Diversity in HIV Epidemic transitions in India: An application of HIV epidemiological metrices and benchmarks

PLOS ONE

***

Response to Reviewers/Editor

We have gone through the observations of the reviewer and editor for our manuscript ID “PONE-D-21-34403R2” titled "Diversity in HIV Epidemic transitions in India: An application of HIV epidemiological metrices and benchmarks". 

We thank the Academic Editor for very constructive comments. We believe the review has enhanced the rigour of the manuscript. 

We have provided a detailed point-by-point response to all observations/comments (editor’s comments in black, our replies in blue). Line numbering refers to the revised manuscript. 

 ***

Academic Editor Observations/Comments and Authors’ Response

1. Comment 1: 

Thanks for adding in uncertainty estimates, they greatly increase the potential usefulness and applicability of the work. Unfortunately, the way they are calculated is not quite correct (based on the way it is described in the methods). It takes into account the uncertainty resulting from the uncertainty in the incidence estimates, but not the uncertainty resulting from the mortality and prevalence estimates. This could mean that the uncertainty is underestimated. Alternatively, if the incidence and mortality/prevalence estimates are correlated between model runs, it may mean that the uncertainty is overestimated.

Reply: 

We agree with the comment. 

2. Comment 2: 

I have not used the Spectrum models myself. Is it possible to export the estimates from individual model runs, using the uncertainty analysis tool? If it is possible, the simplest way to obtain better estimates of the uncertainty would be to divide the incidence by the mortality/prevalence estimate separately for each individual model run, and then to obtain the 2.5% and 97.5% percentiles for the ratios to give 95% uncertainty intervals. Please make sure that the uncertainty intervals that this approach generates seem reasonable to you however, it is possible that the way that the Spectrum model is parameterised and calibrated would mean that this approach would generate unreasonably narrow uncertainty intervals.

Is this is not possible, then there may not be any better approach than the one used, but please discuss the limitations of the way that the uncertainty intervals are generated in the Discussion section.

Reply: 

We thank the Academic Editor for an extremely useful suggestion. The ‘Uncertainty Analysis’ tool in Spectrum indeed generates plausible range by running 1000 iterations. Results of each of the iteration for key indicators may be exported. Given this provision, we revised the method for calculating the uncertainty bounds. The revision has been suitably reflected under Method [Line Number 120-124, 145-153 of revised manuscript], Results [Line Number 170-175, 194-223 of revised manuscript] and Discussions [Line Number 240, 243-245, 249-261 of revised manuscript].

3. Comment 3: 

I agree entirely that the threshold of 0.03 is a rule of thumb only, and feel that a paragraph discussing this is important. What I do not understand is this sentence: “the 0.03 target assumes that secondary infections are equally likely to occur at any time during the (on average) 30 years of time an individual is infected”. As far as I can see, the references you provided in your response say nothing about the target assuming this, and I do not understand how it does. Please explain this statement and/or provide a reference for it, or remove it. It could be replaced with a brief discussion of the major assumption made in setting the threshold: that people live with HIV for a mean of 30 years.

Reply: 

We have revised the paragraph suitably [Line Number 268-272 of revised manuscript].

---

## [Editor Report · Decision Letter 3]

21 Jun 2022

Diversity in HIV Epidemic transitions in India: An application of HIV epidemiological metrices and benchmarks

PONE-D-21-34403R3

Dear Dr. Kumar,

We’re pleased to inform you that your manuscript has been judged scientifically suitable for publication and will be formally accepted for publication once it meets all outstanding technical requirements.

Kind regards,

Nicky McCreesh

Academic Editor

PLOS ONE
---

## [Editor Report · Acceptance letter]

4 Jul 2022

PONE-D-21-34403R3 

Diversity in HIV Epidemic transitions in India: An application of HIV epidemiological metrices and benchmarks 

Dear Dr. Kumar:

I'm pleased to inform you that your manuscript has been deemed suitable for publication in PLOS ONE. Congratulations! Your manuscript is now with our production department. 

Kind regards, 

on behalf of

Dr. Nicky McCreesh 

Academic Editor

PLOS ONE